# Biathletes present repeating patterns of postural control to maintain their balance while shooting

**Justyna Michalska**[1☯]*, **Rafał Zając**[1☯], **Krzysztof Szydło**[1‡], **Dagmara Gerasimuk**[2‡], **Kajetan J. Słomka**[1☯], **Grzegorz Juras**[1☯]

**1** Institute of Sport Sciences, The Jerzy Kukuczka Academy of Physical Education, Katowice, Poland,
**2** Silesian University of Technology – Sports Center, Gliwice, Poland

☯ These authors contributed equally to this work.
‡ KS and DG also contributed equally to this work.
* j.michalska@awf.katowice.pl

**Data Availability Statement:** All relevant data are within the manuscript and its Supporting information files.

## Abstract

Balance can be a main factor contributing to success in many disciplines, and biathlon is a representative example. A more stable posture may be a key factor for shooting scores. The center of foot pressure (COP) is commonly recorded when evaluating postural control. As COP measurements are highly irregular and non-stationary, non-linear deterministic methods, such as entropy, are more appropriate for the analysis of COP displacement. The aim of our study was to investigate whether the longitudinal effects of biathlon training can elicit specific changes in postural control. Eight national-level biathletes, 15 non-athletes who prior to the experiment took part in 3 months of shooting training, and 15 non-athletes with no prior rifle shooting experience took part in our study. The data was collected with the use of a force plate. Participants performed three balance tasks in quiet standing, the shooting position (internal focus–participants concentrated on maintaining the correct body position and rifle), and aiming at the target (external focus–participants concentrated on keeping the laser beam centered on the targets). Biathletes obtained significantly lower values of sample entropy compared to the other groups during the shooting and aiming at the target trials ($p<0.05$). External and internal focuses influenced the process of postural control among participants who had prior rifle shooting experience and the control group; they obtained significantly higher values of sample entropy while shooting and aiming at the target compared to the quiet standing trial ($p<0.05$). The biathletes obtained significantly lower values of sample entropy in the aiming at the target position compared to the quiet standing trial. Specific balance training is associated with the ability to deal with a more challenging, non-specific task. The biathletes seemed to employ a different motor control strategy than the beginners and control group, creating repeating patterns (more regular signal for COP) to keep one's balance during the shooting and aiming at the target positions.

**Funding:** The authors received no specific funding for this work.

**Competing interests:** The authors have declared that no competing interests exist.

## Introduction

Coaches in most sports disciplines must face many challenges involving the physical preparation of their athletes. Among them, overloaded calendars and, now more than ever, limited access to sports facilities seem to be especially important. The proper management of athletes' health should be considered a priority. Developing better postural control can be an effective way to achieve this goal [1]. Balance can also be a main factor contributing to success in many disciplines, and biathlon is a representative example [2]. The competition is divided into two parts (skiing and shooting), each demanding a specific set of skills. Long-distance skiing requires a significant endurance capacity. Moreover, maximal physical effort influences postural balance and rifle stability during aiming [3]. To perform well in shooting, an athlete needs to be concentrated, precise, and maintain their balance while facing significant physiological stress. Recent studies have shown that elite female and male shooters are characterized by less body sway than non-elite shooters [4–6]. Additionally, significant differences in this ability can also be observed between high-level and low-level biathletes [4, 7, 8]. The center of foot pressure (COP) is commonly recorded when evaluating postural control. Because COP measurements are highly irregular and non-stationary, non-linear deterministic methods are more appropriate for the analysis of COP displacement [9]. They allow for the exploration of the randomness or predictability of COP fluctuations [10, 11]. One of the frequently-used methods to measure the complexity and regularity of COP signals is entropy [12].

Although entropy has been used to examine the complexity of human postural control over the last two decades, limitations remain with regard to our understanding around complexity [13]. In general, entropy describes the regularity of a time series. Regularity quantifies the unpredictability level of the COP fluctuation over time by analyzing the probability of the repetition of a particular sequence of COP values over time. Low values indicate a more regular signal (pathology/disability), and high values are a sign of more chaotic signal properties (higher expertise) [14, 15]. This method of interpretation can suggest that the postural control system response to an unexpected perturbation has a specific pattern that can be observed by measuring the entropy of the signal [16–18]. Manor et al. [19] observed that the degree of complexity associated with postural control system behavior was correlated with the integrity of the involved sensory systems. Somatosensory and/or visual impairments contributed to decreased postural sway complexity, which reflected the reduced adaptive capacity of the postural control system.

Not all studies support the above interpretation. Some authors have observed that older adults show a higher degree of complexity (higher entropy) compared to younger adults [12, 20, 21]. This result might indicate impaired sensory systems which fail to provide a precise input for postural control. Taylor et al. [22] similarly noticed that patients with dementia (DP) were characterized by higher values of sample entropy (SampEn) during gait trials.

On the other hand, some results contradict the hypothesized higher degrees of complexity in athletes not only in clinical practice, but also in sports. There are studies that showed no changes in the regularity of COP between a highly-skilled group and a healthy age-matched control group (CG) during a simple balance task [23–25]. In turn, Schmit et al. [26] observed that ballet dancers have decreased entropy compared to runners, even though dancers are considered more proficient in using their balance ability. Additionally, Raffalt et al. [27] showed that rifle shooters had a lower entropy than non-shooters during a single leg stance. With such a broad range of results, it is difficult to clearly determine if a higher or lower entropy value can be attributed to postural control efficiency. Rhea et al. [28] suggested that before any conclusions are made, it is necessary to first take a look at the direction of changes and try to find an explanation for those changes.

Roerdink et al. [29] noticed a direct relation between the amount of attention invested in postural control and the regularity of COP fluctuation. More irregular COP fluctuations (as indexed by an increase in sample entropy) may be interpreted as an increase in the efficiency or 'automaticity.' It is consistent with the 'constrained action hypothesis' proposed by Wulf [30], which demonstrates the presence of automatic motor control processes when attention is withdrawn from controlling one's balance. The attention can be experimentally diverted from posture when attentional focus is directed to the effects of one's movement in the environment (external focus–EF) as compared to when one's focus is directed to body movements (internal focus–IF). Research has shown that EF is more beneficial for motor performance and learning (including balance) relative to IF [30]. EF facilitates movement efficiency by promoting movement organization at a more automatic level, while IF involves a more conscious control of effectors and consequently disrupts the automaticity of coordination processes.

The aim of the present study was to investigate the regularity of the COP trajectory of biathletes in positions closely related to the sports competition (shooting position–IF, and aiming for the target–EF). To examine whether the longitudinal effects of biathlon training can elicit specific changes in postural control, two research groups apart from professional biathletes were recruited: non-athletes, who completed 3 months of shooting training, and a young, healthy CG. The authors hypothesized that biathletes would show a more regular COP signal compared to the other groups. The authors also assumed that an EF would induce the biggest changes in the automaticity of postural control.

## Materials and methods

### Participants

Thirty-eight healthy, young participants (32 males, 6 females) took part in the experiment. The first group consisted of 8 national-level biathletes. The second was formed by 15 non-athletes who, prior to the experiment, took part in 3 months of shooting training, including two practice sessions per week (referred to as "beginners"). The control group (CG) consisted of 15 non-athletes with no prior rifle experience. A detailed description of all group characteristics is presented in Table 1. Participants were excluded if they presented a lower or upper limb injury at the time of the experiment or one month prior. The participants gave a written informed consent for their voluntary participation in the study. The study was approved by the Institutional Bioethics Committee.

### Experimental procedure

Participants performed three balance tasks while standing barefoot on a force plate. First, they were asked to stand in a comfortable position with their feet approximately at shoulder-width distance on the force platform, with their gaze fixated at a reference point located 5m away in front of them (QS–quiet standing). Next, they were asked to stand while holding a rifle in a standing shooting position without any fixation point in front of them (SP–shooting position). They were asked to focus on holding this shooting posture. In the last trial, participants were

**Table 1. Characteristics of the experimental groups.**

|  | Control | Beginners | Biathlon |
|---|---|---|---|
| N | 15 | 15 | 8 |
| Age [years] | 21.2 ± 0.8 | 22.3 ± 0.6 | 26.1 ± 5 |
| Height [cm] | 180.5 ± 7.8 | 179.6 ± 6.5 | 170.6 ± 9.5 |
| Weight [kg] | 74.8 ± 9.5 | 73.9 ± 8.8 | 58.9 ± 11.5 |

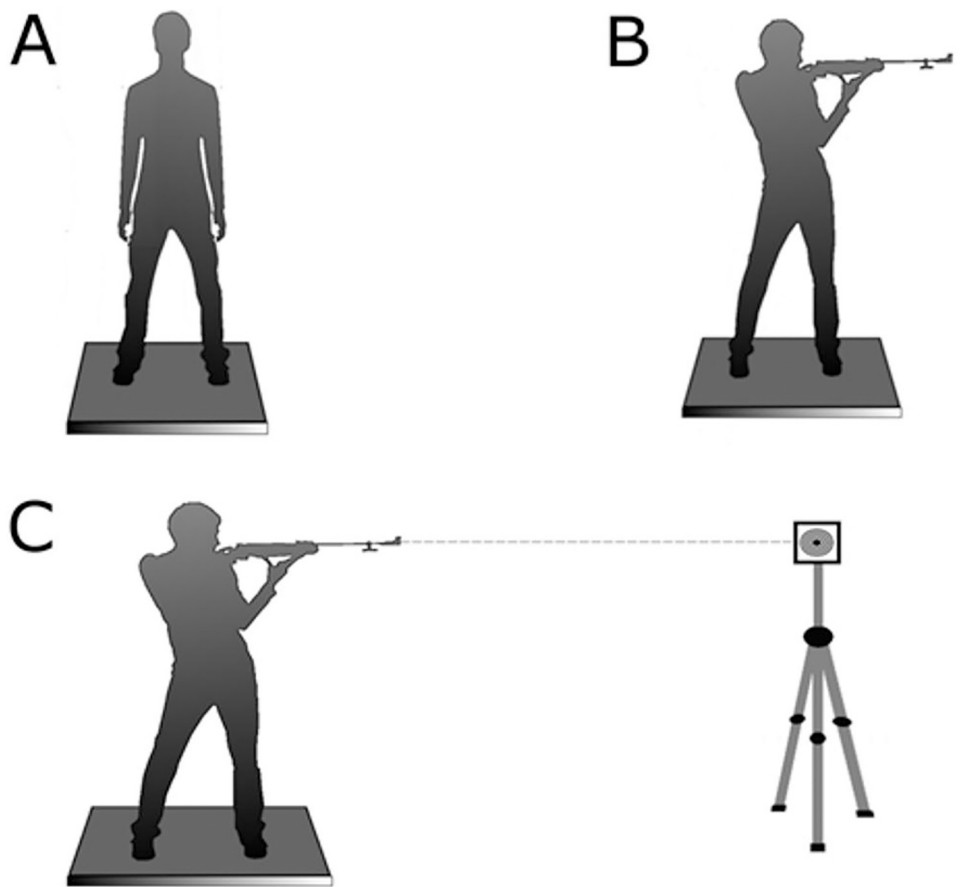

**Fig 1. Experimental procedure.** A–quiet standing; B–shooting position; C–aiming at target.

given a target at a distance of 5 meters (the target size was adjusted for distance), and were asked to focus on aiming and holding a laser mark inside the target (AT–aiming at target). The laser pointer was fixed at the end of the rifle's barrel (Fig 1).

Participants in the CG were instructed before the examination about the proper technique of the biathlon shooting position i.e., feet slightly wider than hip-width apart, with feet positioned 90 degrees to the target and body weight evenly distributed on both legs. To provide a stable rifle position, participants were instructed to align the position of the hip and elbow to be directly under the rifle.

Each trial lasted 30 seconds and was repeated three times. The order of the trials was randomized for each participant.

## Data analysis

The data was recorded with the use of the AMTI BP-600900 force platform and AMTI Netforce software. The sampling frequency was set to 50 Hz. The first step of data processing included the low-pass filtering of each measurement using a 4th order Butterworth filter with a cut-off frequency of 7 Hz. The next step involved calculating the center of pressure from the ground reaction forces (Fx, Fy, Fz) and moments (Mx, My, Mz), which constituted the basis for the evaluation of SampEn of the signal. The CoP displacement increment time series was used in the sample entropy analysis. Sample entropy is defined as the negative logarithm for

the conditional probability that vectors of length m, which are similar in a time series, remain similar for length m+1 (excluding self matches). Calculations were based on the method described by Richman and Moorman [14] using the following formula in Eq (1):

$$SampEn(m, r, N) = -\log\left(\frac{A^{m+1}(r)}{B^m(r)}\right) \qquad (1)$$

where m represents the number of data points to compare in a data series, r represents the tolerance level, and N represents the length of the time series.

A is the number of alike vector lengths (m+1) falling within a relative tolerance limit (r times the standard deviation of the time series), and B is the number of alike vector lengths (m) falling within the tolerance limit.

A perfectly repeatable time series (with similar distances between data points) would elicit a SampEn value equaling 0, and a perfectly random time series would give a SampEn value converging toward infinity [31]. The values of the input parameters in this experiment were based on the results of Ramdani et al. [11]. The template vector length (m) was set to 3 and tolerance (r) to 0.2. All data processing was performed using Python (Python Software Foundation. Python Language Reference, version 3.7. Available at http://www.python.org).

## Statistical analysis

Before conducting statistical analyses, repeated measurements from each trial were averaged across all participants. Then normality of the data distribution was confirmed by the Shapiro-Wilk test. The homogeneity of the variances between groups was evaluated with Levene's test and within participant groups by Mauchly's test of sphericity. To evaluate the effect of the group and task on SampEn, a two-way mixed-design ANOVA with the group (between participants) and task (within participants) as independent factors was applied to the dependent variable in each direction (AP–anterior posterior, ML–medio lateral). With regard to the overall effect of a group, task, or interaction between two factors, pairwise t-test comparisons with Bonferroni corrections were applied. As a means of measuring effect size, generalized eta square statistics were used [32]. Effect sizes were evaluated as small (0.01), medium (0.06), and large (above 0.14) based on the guidelines described by Cohen [33]. The level of significance was set to 0.05. All statistical analyses were performed in R (R Core Team (2020). R: A language and environment for statistical computing. R Foundation for Statistical Computing, Vienna, Austria. URL https://www.r-project.org/).

## Results

### Sample entropy in the anterior-posterior plane

**The effect of group.**   Our results showed a significant main effect of group (biathletes, beginners, and CG) on SampEn (F = 10.422; df = 2; p<0.001; $\eta_G^2$ = 0.272). Post hoc tests showed no differences between groups in QS (p>0.05). There was a significant difference in SP between the CG and biathletes (p< 0.001) and between beginners and biathletes (p<0.001). The same pattern was observed in the AT condition, where there was a significant difference between the CG and biathletes (p<0.01) and between beginners and biathletes, (p<0.001) (Fig 2). In both trials there was no difference between controls and beginners (p>0.05).

**The effect of task.**   A significant main effect of task (QS, SP, AT) was confirmed for SampEn (F = 7.998; df = 2; p<0.001; $\eta_G^2$ = 0.079). As expected, post hoc tests demonstrated that in the CG, SampEn differed significantly in SP and AT when compared to QS (p<0.001), but no difference was observed between SP and AT trials (p<0.001). Similarly, for the beginners

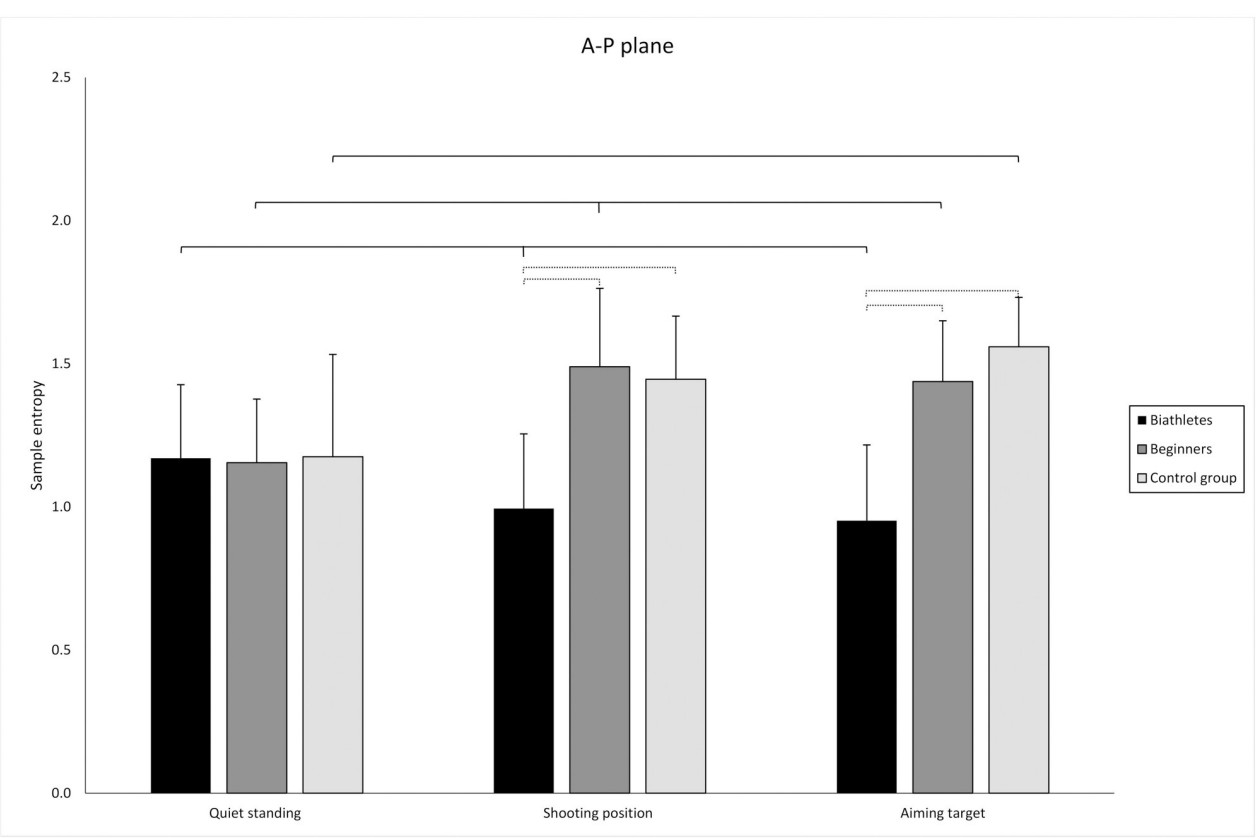

**Fig 2. Mean values of SampEn in the anterior-posterior direction (standard deviations marked as error bars) in the biathlete, beginner, and control groups.** Thin dotted line: significant differences between groups. thin solid line: significant differences between conditions among groups.

SampEn was significantly different in SP and AT when compared to QS ($p < 0.001$), but no differences were observed between SP and AT conditions ($p > 0.05$). In the biathletes group, no significant change in SampEn was observed between QS and SP and between SP and AT. The only significant difference for this group was observed between QS and AT trials, $p = 0.04$ (Fig 2). It is worth mentioning that in the biathletes group, the changes between consecutive conditions were much smaller and, contrary to the values for both other groups in the SP and AT trials, were lower than in the QS condition.

**Group*task interaction.** There was a significant main group-task interaction on SampEn ($F = 9.909$, df = 4; $p < 0.001$, $\eta_G^2 = 0.174$).

## Sample entropy in the medio-lateral plane

**The effect of group.** The main effect of group (biathletes, beginners, and CG) was significant for SampEn ($F = 24.167$; df = 2; $p < 0.001$; $\eta_G^2 = 0.365$). Post hoc tests showed no differences between groups in QS ($p > 0.05$). There was a significant difference in SP between the CG and biathletes ($p < 0.001$) and between beginners and biathletes ($p < 0.001$) (Fig 3). The same pattern was observed as in the AT condition. In particular, there was a significant difference between the CG and biathletes ($p < 0.05$) and between beginners and biathletes ($p < 0.001$) (Fig 3). In both trials (SP and AT), there was no difference between the control and beginner groups ($p > 0.05$).

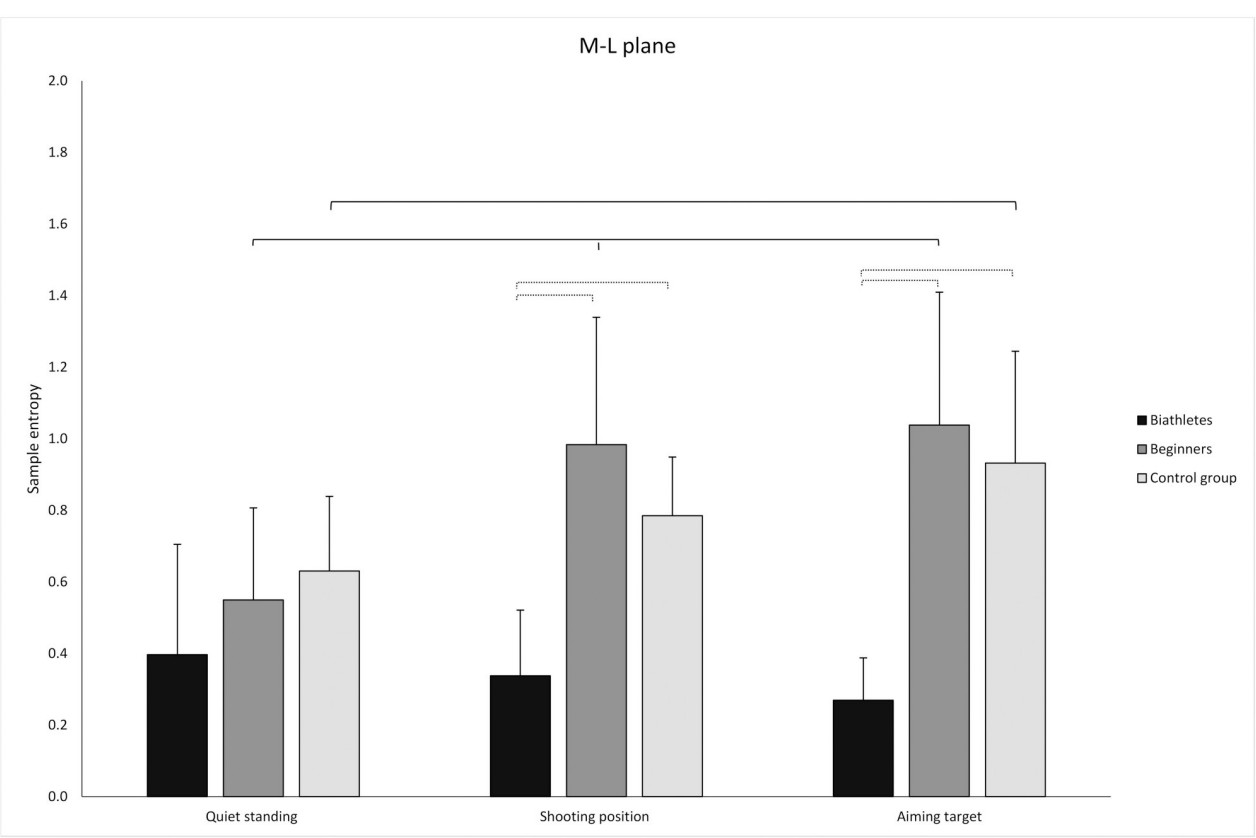

**Fig 3. Mean values of SampEn in the medio-lateral direction (standard deviations marked as error bars) in the biathlete, beginner, and control groups.** Thin dotted line: significant differences between groups. thin solid line: significant differences between conditions among groups.

**The effect of task.** Our results showed significant main effect of task (QS, SP, AT) on SampEn (F = 7.169; df = 2; p = 0.001; $\eta_G^2$ = 0.107). Post hoc tests demonstrated that in the CG only a significant difference in the ML direction was observed between the QS and AT conditions (p<0.001) (Fig 3). However, for the beginners the changes were similar to the AP direction with significant differences observed between the QS and SP (p<0.001) and QS and AT conditions (p<0.001) (Fig 3). There were no significant differences between experimental conditions in the biathletes (p>0.05).

**Group\*task interaction.** There was a significant main group-task interaction on SampEn (F = 4.525; df = 4; p = 0.003, $\eta_G^2$ = 0.131).

## Discussion

The aim of the present study was to investigate whether an internal and external focus of attention could promote greater changes in the postural regularity of COP. We also aimed to explore how the process of postural control differs between groups of different levels of expertise. Our results supported the hypothesis that biathletes would show a more regular COP signal compared to other groups. However, the second hypothesis could not be supported, because EF did not induce the biggest changes in the automaticity of postural control compared to IF.

In the present study, there were no significant differences in the regularity of COP between the three research groups during QS in both the AP and ML planes. This result is in line with

other studies investigating the regularity of COP among athletes in trials which included a dual task condition [23], a more specific posture like a ballet position [24], or more demanding balance conditions like eyes closed or an unstable surface [25]. In all mentioned works, the experimental conditions led to an increase in entropy among athletes compared to the CG. Maintaining an upright body posture is quite easy for all healthy participants. Therefore, we assumed that professional sports training does not transfer balance improvements to simple motor tasks, or QS does not require "elite level" balance.

Biathletes were characterized by lower values of entropy, especially during the SP and AT trials and both in the AP and ML plane, compared to the other two groups, which confirms our first research hypothesis. This result stands in contrast with the research cited above, where an additional or more demanding balance task led to a decrease in COP regularity among athletes. However, we must consider a few important aspects of this examination. First, we cannot ignore the fact that for professional biathletes, SP and AT trials are a familiar position. Years of professional training made it more automatic and required less attentional resources. Therefore, further tasks should not interfere significantly (or at least significantly less than untrained groups) with the main motor task. This claim corresponds well with Fitts's *Motor Learning Theory* [34]. According to the theory, after many years of sports training, biathletes reached the third (autonomous) stage of motor learning. It can be also observed in our results, specifically in the increased regularity of the COP signal during AT trials compared to QS and SP in the AP plane, as well as no significant changes between trials in the ML plane. There is strong evidence that biathletes present different postural control strategies, which ensures motor success. These outcomes are contrary to the CG and BG results, which are in line with the "*constrained action hypothesis.*" For BG and especially for CG, SP and AT were not familiar positions. In contrast to the biathletes, we could distinguish both a primary (balance) and secondary task (IF and EF) in these motor tasks. Introducing additional focus in BG and CG experimentally withdrew attention from the balance task; this was shown by higher values of sample entropy, which might correspond with the more automatic process of the primary motor task. According to Rhae et al. [28], EF provide bigger changes in values of sample entropy compared to IF. However, in our study there were no differences between IF and EF. The two types of additional focus included in this research triggered the same changes in COP structure, which does not confirm our second research hypothesis.

To achieve good results, elite biathletes aim at decreasing postural and rifle sway in the SP [4]. Performing a shot as soon as a stable position is obtained is a crucial factor for success [35]. Efforts put towards following this strategy can be observed in the experimental data. Postural sway and the regularity of the COP signal tend to be positively correlated [36]. This was also confirmed by the research by Raffalt et al., [27] who observed that sporting rifle shooters are characterized by a higher regularity of their COP signal and lower body sway compared to non-shooters during single-legged standing. The same pattern of changes in the measured data can be also observed in our research; biathletes were characterized by a significantly more regular COP signal in positions closely related to their competition.

When competing, biathletes perform in quasi constant conditions. They use the same sportswear as during their training, the target is at a familiar distance, and they shoot only after they adjust to the environmental conditions and when they can stabilize their posture. In our study, this is even more clear in the SP and especially the AT conditions, as aiming was not followed by a shot. A main goal in order to achieve success in shooting is to minimize the movements of the gun barrel. This, however, requires a high level of postural stability [6]. In addition, movement variability in a performed task diminishes in the process of professional training [37]. Therefore, we can also expect that in experimental conditions biathletes will perform significantly better in this aspect compared to other groups. In the present study, as

expected, biathletes were characterized by significantly lower values of SampEn during AT compared to QS. We can therefore conclude that their level of proficiency is reflected to some extent in our data. Another conclusion could be that well-developed postural control mechanisms may play a significant role in achieving success in competition.

Based on the current research, biathletes are characterized by a more regular signal of COP. This means that they have a more predictable, repetitive pattern of postural control. It is speculative whether additional disturbing external factors should be introduced into the training regimen in order to push this limit a little further. There is a chance that the athlete, knocked out of their comfort zone, will perform worse due to the disturbance in their repetitive and known patterns. It seems to be enough that they have to deal with fatigue and changing weather conditions. It is worth pointing out that success in shooting is dependent on barrel variability and may not be as related to the body oscillations [38]. Other aspects influencing the final result, e.g. the size of the sight, distance to the target, shooting equipment and position, etc., are constant. In the author's opinion, one the aims of training is to develop a postural control pattern that ensures the athletes' sports success, and overstimulation might decrease their shooting accuracy. We can certainly monitor progress by observing the changes in signal entropy over time, such as during a training program, which provides valuable information for an athlete's progress.

This study presents several few limitations. The main limitation is that biathletes begin shooting in a heavily fatigued state after covering a certain cross-country distance. In our research, they performed a balance examination at full rest. Secondly, during their competition biathletes perform a shot after acquiring a good position, when in our study they had to stay in a position concentrated on the target for the desired time. Third, the "aiming at a target" condition was essentially a dual-task condition; however, we did not record performance with regard to aiming at a target in conjunction with postural sway. We recommend that future studies should include a record of how long the subject has been aiming at the target. Lastly, our results for the beginners were closely related to the CG, which suggests that three months of training is not sufficient for inducing changes in the postural control system.

## Conclusion

The results of the present study suggest that specific balance training is associated with the ability to deal with a more challenging nonspecific task such as SP and AT. Furthermore, biathletes seem to employ a different motor control strategy than beginners and participants in the CG. Among biathletes, the variability of the COP signal was significantly lower than in the other groups. This contributes to a more stable posture for this group, which is essential for producing a successful shot. Our results do not support the theory that connects lower SampEn values with poor balance or postural control deficiency.

## Supporting information

**S1 Data.**
(XLSX)

## Author Contributions

**Conceptualization:** Justyna Michalska, Dagmara Gerasimuk, Kajetan J. Słomka, Grzegorz Juras.

**Data curation:** Justyna Michalska, Rafał Zając, Krzysztof Szydło, Dagmara Gerasimuk.

Formal analysis: Justyna Michalska, Rafał Zając.

Investigation: Justyna Michalska, Rafał Zając, Krzysztof Szydło.

Methodology: Justyna Michalska, Rafał Zając, Krzysztof Szydło, Kajetan J. Słomka.

Project administration: Dagmara Gerasimuk, Kajetan J. Słomka, Grzegorz Juras.

Resources: Justyna Michalska, Rafał Zając, Krzysztof Szydło.

Software: Rafał Zając.

Supervision: Dagmara Gerasimuk, Grzegorz Juras.

Validation: Justyna Michalska, Rafał Zając, Kajetan J. Słomka.

Visualization: Justyna Michalska.

Writing – original draft: Justyna Michalska, Kajetan J. Słomka.

Writing – review & editing: Justyna Michalska.

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
