## [Decision Letter · Decision Letter 0]

15 Dec 2021

PONE-D-21-34186Biathletes prefer a closed-looped strategy to maintain their balance during shootingPLOS ONE

Dear Dr. Michalska,

Thank you for submitting your manuscript to PLOS ONE. After careful consideration, we feel that it has merit but does not fully meet PLOS ONE’s publication criteria as it currently stands. Therefore, we invite you to submit a revised version of the manuscript that addresses the points raised during the review process.

The Reviewers have made numerous detailed suggests for changes that can strengthen the manuscript. I have classified these changes as a Major Revision because I think they are worthwhile (and worth doing carefully). I agree with the Reviewers that the requested changes will significantly enhance the value of your contribution.

We look forward to receiving your revised manuscript.

Kind regards,

Thomas A Stoffregen, PhD

Academic Editor

PLOS ONE

Journal Requirements:

Reviewers' comments:

Reviewer's Responses to Questions

**Comments to the Author**

1. Is the manuscript technically sound, and do the data support the conclusions?

Reviewer #1: Yes

Reviewer #2: Partly

2. Has the statistical analysis been performed appropriately and rigorously? 

Reviewer #1: Yes

Reviewer #2: No

3. Have the authors made all data underlying the findings in their manuscript fully available?

Reviewer #1: Yes

Reviewer #2: Yes

4. Is the manuscript presented in an intelligible fashion and written in standard English?

Reviewer #1: Yes

Reviewer #2: No

5. Review Comments to the Author

Reviewer #1: General: This paper exampled postural sway (specifically sample entropy) in biathletes relative to beginners and a control group. The study design allows for interpretation about how skill level and task constraints may influence the control of posture. While the manuscript is generally in good shape, I offer a few revisions below to help strengthen it in a few places.

Comments and suggestions to the authors:

ABSTRACT

Line 12: The word “also” can be removed, as no other concepts have been discussed at this point.

INTRODUCTION

Lines 37-55: The beginning of this paragraph discusses injury risk and cites literature in that area (lines 37-42). However, the rest of the paragraph focuses on task/skill performance, not injury risk. Since the rest of your paper focuses on performance, I recommend removing the anchoring to injury risk at the beginning of this paragraph and keep the paper solely focused on performance rather than injury risk. Conceptually, that will be a stronger connection to your study design, as no one is at risk of injury from the balance tasks you uses, but their performance could (and did) vary among the tasks.

Line 57: The reference below would support the first sentence in your paragraph.

Rhea, C. K., Silver, T. A., Hong, S. L., Ryu, J. H., Studenka, B. E., Hughes, C. M., & Haddad, J. M. (2011). Noise and complexity in human postural control: interpreting the different estimations of entropy. PLOS ONE, 6(3), e17696.

Line 86: I recommend removing the word “recent”, as the cited review paper is 8 years old and the studies within that review paper are even older.

METHODS

Line 106: Define CG before using the acronym.

Line 112 and/or 171: Please report whether there were any significant differences between the groups with respect to age, height, and weight. It looks like the biathlon group was smaller with respect to height and weight. BMI has been shown to affect postural control.

Ku, P. X., Osman, N. A., Yusof, A., & Abas, W. W. (2012). Biomechanical evaluation of the relationship between postural control and body mass index. Journal of biomechanics, 45(9), 1638-1642.

Lines 116 and 129: In line 116, it states that participants were asked to “stand in a comfortable position with their feet approximately at shoulder width distance”, but line 129 says “Participants in the CG were instructed before examination about the proper technique for the standing SP i.e., feet slightly wider than hip width, with feet positioned 90 degrees to the target and body weight evenly distributed on both legs”. Did the control group receive different instructions than the other groups? If so, why?

Line 139: Was the CoP displacement time series used in the sample entropy analysis (relative to the velocity or an increment time series)? If so, please state that here.

DISCUSSION

Line 222: Please add a sentence at the end of this paragraph that states whether your hypotheses were supported and to what extend (i.e., generally, partially, not at all).

Line 287: An additional limitation should be considered. The “aiming to a target” condition was essentially a dual-task condition, so it would have been ideal to also record performance on target aim in conjunction with postural sway. While you may not have these data, it would be good to identify has a limitation or at least a future direction.

General comment: The discussion primarily focuses on how the biathletes differed from the other groups, which is appropriate given the research question. However, it would be helpful to add text to describe/interpret the findings of the beginners relative to the control group. Significant differences are reported between those groups in Figures 2 and 3, so describing why those differences may have occurred would be important here.

Reviewer #2: PONE-D-21-34186review

PLOS One

The authors examine complexity changes within a specific population group that has extensive balance experience relative to a shooting/aiming task and make comparisons of balance performance to a novice and control group. Using sample entropy the COP complexity is indexed for signal regularity and quantified for the 3 different conditions (quiet standing, shooting, and aiming). The main findings show that the biathletes exhibit different COP complexity than the novice and control groups.

Within the current form of the manuscript, the authors have an appropriate experimental design and sample to address the main question. However, there lacks adequate rationale and clarity on the background theory and evidence related to this question. The authors need to make significant improvements around the main concepts (balance experience, complexity, attentional focus) and provide clear links of these ideas to their main question/variables.

Title – this does not appear to align with information presented within the manuscript and the concepts provided in the introduction. Closed-looped strategy is not explained/described anywhere within the manuscript. Please address.

Abstract

p. 8. Line 12 – the abstract starts with a phrasing that uses “also” in the first sentence. I don’t see how this fits within the first sentence. Also, the term “success” has a quite a few different interpretations/meanings. It would be beneficial to clarify on how the authors interpret this term.

p. 8 line 16 – I would recommend moving “such as entropy” to after deterministic methods.

p. 8 line 17 – At the beginning of sentences numbers need to be spelled out as do numbers less than 10.

Data is plural – should read “The data were collected…”

The conditions related to internal/external focus should be incorporated into the background information of the abstract.

Some additional details around the methods would be helpful. This could be achieved by removing the details of the non-significant findings in the results part.

The final sentence includes “closed-looped strategy” – see comment about title as that applies here in the abstract too.

Introduction

The opening paragraph covers two main ideas that should be separated into independent paragraphs. First, the role of balance on performance and links to injury risk followed by the explanation of COP measurement. Also, the authors are making a general claim on injury risk in relation to balance/postural control. There needs to be improved rationale on this element (if this is a focus of the paper).

Line 56-57: The statement “its interpretation is still intangible” is not accurate. Significant work has been done on this front and while it is recognized that there remain limitations to our understanding around complexity. Concrete hypotheses associated with this characteristic of human behavior can be found within the literature. See works referenced below (in addition to the currently used reference with Lipsitz).

• Manor, B., Costa, M. D., Hu, K., Newton, E., Starobinets, O., Kang, H. G., Peng, C. K., Novak, V., & Lipsitz, L. A. (2010). Physiological complexity and system adaptability: evidence from postural control dynamics of older adults. Journal of Applied Physiology, 109(6), 1786–1791. https://doi.org/10.1152/japplphysiol.00390.2010

• Vaillancourt, D.E., & Newell, K.M. (2003). Aging and the time and frequency structure of force output variability. Journal of Applied Physiology, 94, 903–912. doi:10.1152/japplphysiol.00166.2002

The current manuscript would be greatly strengthened when the authors incorporate these perspectives within the introduction. This also will provide them with clearer interpretations of the findings based on the experimental design. Particularly, given the somewhat bi-directional changes in entropy between the experimental groups it is worthwhile to address in more detail how the balance training/experience in biathletes influences the reduction of COP irregularity.

Page 11, lines 94-98

o These two sentences do not align because one is looking at sway magnitude (which is not used in the methods/results section) while the other predicts COP regularity. Clarify whether you expected directional changes for each type of COP variability metric (linear/nonlinear).

o New term introduced here that needs context and background information. I am assuming it relates to the constrained action hypothesis but that is because I have familiarity with this perspective whereas other readers may not. Thus, novel readers need more context and explanation on what you mean by automaticity.

Methods

o Did participants wear shoes?

o Do you know whether participants fixated on a spot during the shooting position condition? Is there additional details from your instructions or did you conduct a commitment check about their focus? The commitment check would also apply to the aiming condition since I think you are tagging that as the external focus of attention manipulation.

Line 138-139: the calculation of COP description is not clear. The AMTI force plate will provide a direct output of the COP trajectory without the need for calculation. Can you please further explain this sentence.

Improvement in the description of the Sample Entropy analysis is needed to help readers not fully familiar with this non-linear metric – particularly, lines 139-142.

Results

The authors need to check the degrees of freedom for the ANOVA results. Based on the design, there are 3 groups and 3 tasks with a total of 38 participants included (unless some were omitted but that would need to be added to the manuscript if that was the case), this would lead to different DOF in the F statistic.

o It would be beneficial to reference figures 2 and 3 earlier in the results section to help draw the reader’s attention to the significant differences. I know the figures will be placed nearby but the first paragraph of the AP and ML results relate very much to the data presented in these figures.

p.15 line 177-179 – I think these two sentences need to be merged together do better illustrate that the group differences were in the AT condition.

For the A-P direction results, the authors provide the significant interaction effect but then I am confused by the remaining details in the first paragraph compared to the second. If an interaction effect is present there is no need to follow-up on the main effects (group and condition), but it is hard to follow whether second paragraph (lines 181-193) is the post-hoc analysis of the significant interaction effect.

o This also exists in the M-L direction results. Please address.

p. 15 line 188-189: How do you know there was a small change between consecutive trials when the data were averaged across the trials as stated in the statistical analysis section? Please clarify and/or adjust the manuscript.

Discussion

p.17, line 220 – attentional focus has not been discussed or included in the original hypotheses. If you are going to interpret the findings relative to this concept additional background information is needed in the introduction.

p. 17, line 223-231 – There appears to be some inconsistencies in the information detailed in this paragraph. The first sentence states no differences between groups, followed by that being in line with other studies. But, then it states increased entropy for athletes. Please clarify.

p. 18, line 245-246 – The reference to the findings being in line with the “constrained action hypothesis” needs further elaboration. In what way do the results support the many elements of this perspective?

p. 18, line 263-264 – the interpretation of close-loop (should be closed-loop) pattern needs further explanation on why the authors believe this to be the case within the results. Plus, as with previous comments around this concept, a rationale for its use is needed in the introduction.

p. 19, line 282-286 – depends on the task demands, which authors are correct here in terms of reducing COP variability irregularity but other tasks may not follow the same pattern.

Also, task success in shooting is dependent on the barrel variability and may not be as related to the movement/body variations. Classic work by Arutyunyan showed this characteristic.

o Arutyunyan GA, Gurfinkel VS, Mirskii ML. Organization of movements

on execution by man of an exact postural task. Biophysics. 1969; 14:1162Y7.

Minor Comments

Table 1 needs a period instead of a comma under the Biathlon data

p. 16, line 199 – there is only one interaction so it should be used in the plural form – should read “significant interaction.”

p. 18, line 254 – please address this sentence for clarity purposes.

p.19, line275 – biathletes should be in lower case for consistency within the manuscript

6. PLOS authors have the option to publish the peer review history of their article (what does this mean?). If published, this will include your full peer review and any attached files.

Reviewer #1: **Yes: **Christopher K Rhea

Reviewer #2: No

---

## [Author Response · Author response to Decision Letter 0]

27 Jan 2022

Reviewer #1

General: This paper exampled postural sway (specifically sample entropy) in biathletes relative to beginners and a control group. The study design allows for interpretation about how skill level and task constraints may influence the control of posture. While the manuscript is generally in good shape, I offer a few revisions below to help strengthen it in a few places.

Answer: 

First of all, we would like to express thanks to the reviewer for a very insightful review. The manuscript was corrected according to the reviewer’s suggestions. Your comments have contributed to improvement our paper.

Reviewer comment

ABSTRACT Line 12: The word “also” can be removed, as no other concepts have been discussed at this point.

Answer:

Thank you for this comment. We have removed the word “also” from the abstract. 

Reviewer comment

INTRODUCTION Lines 37-55: The beginning of this paragraph discusses injury risk and cites literature in that area (lines 37-42). However, the rest of the paragraph focuses on task/skill performance, not injury risk. Since the rest of your paper focuses on performance, I recommend removing the anchoring to injury risk at the beginning of this paragraph and keep the paper solely focused on performance rather than injury risk. Conceptually, that will be a stronger connection to your study design, as no one is at risk of injury from the balance tasks you uses, but their performance could (and did) vary among the tasks.

Answer:

We agree with the reviewer, that we did not focus on the risk of injury, so we have removed the sentence from the introduction: “Together, those factors significantly increase the risk of injury”. We have also removed the first citation.

Reviewer comment

Line 57: The reference below would support the first sentence in your paragraph. Rhea, C. K., Silver, T. A., Hong, S. L., Ryu, J. H., Studenka, B. E., Hughes, C. M., & Haddad, J. M. (2011). Noise and complexity in human postural control: interpreting the different estimations of entropy. PLOS ONE, 6(3), e17696.

Answer:

Thank you for pointing this out. We have added this reference, following the reviewer’s suggestion. 

Reviewer comment

Line 86: I recommend removing the word “recent”, as the cited review paper is 8 years old and the studies within that review paper are even older.

Answer:

We do agree with reviewer comment, therefore the sentence was changed from “Recent studies have shown that an external focus (EF) is more beneficial…” to “Review paper have shown that an external focus (EF) is more beneficial…”

Reviewer comment

METHODS Line 106: Define CG before using the acronym

Answer:

The acronym CG was used for the first time in the introduction and was defined as a control group (line 73 in the original manuscript). However, following the reviewers comment, we have introduced this acronym in the method in the revised manuscript.

Reviewer comment

Line 112 and/or 171: Please report whether there were any significant differences between the groups with respect to age, height, and weight. It looks like the biathlon group was smaller with respect to height and weight. BMI has been shown to affect postural control. Ku, P. X., Osman, N. A., Yusof, A., & Abas, W. W. (2012). Biomechanical evaluation of the relationship between postural control and body mass index. Journal of biomechanics, 45(9), 1638-1642

Answer:

Thank you for this comment. We are aware of the differences between biathletes and other groups. However, years of training change the proportions of the body, in particular, the ratio between adipose tissue and muscle tissue. It is difficult to select similar control group characterized by the same antropometric parameters. Additionally, we have checked the correlation between BMI and variables of SampEn. See bellows:

variables Pearson correlation AP plane 

p < ,05000

 Group SampEn - QS SampEn - SP SampEn - AT

BMI Beginner 0.06489 0.45594 0.22655

BMI Biathlon -0.13933 0.19950 -0.16900

BMI Control 0.00071 -0.15276 -0.19767

variables Pearson correlation ML plane 

p < ,05000

 Group SampEn - QS SampEn - SP SampEn - AT

BMI Beginner 0.36622 -0.15839 -0.06615

BMI Biathlon 0.04901 0.07496 -0.00655

BMI Control 0.03593 -0.13647 -0.48594

There was no significant correlation between BMI and values of SampEn, neither in AP and ML plane.

Reviewer comment

Lines 116 and 129: In line 116, it states that participants were asked to “stand in a comfortable position with their feet approximately at shoulder width distance”, but line 129 says “Participants in the CG were instructed before examination about the proper technique for the standing SP i.e., feet slightly wider than hip width, with feet positioned 90 degrees to the target and body weight evenly distributed on both legs”. Did the control group receive different instructions than the other groups? If so, why?

Answer:

We fully understand reviewer’s concern, however all subjects received the same instruction. First instruction “…they were asked to stand in a comfortable position with their feet approximately at shoulder width distance on a force platform…” corresponds to quiet standing task, while the second instruction “…feet slightly wider than hip width, with feet positioned 90 degrees to the target and body weight evenly distributed on both legs…” corresponds to shooting position (SP). It was necessary to explained this position to the control groups, because they were not familiar with it, unlike beginners and biathletes. For better understanding we have corrected the sentences as follow: “Participants in the CG were instructed before examination about the proper technique of the biathletes shooting position”.

Reviewer comment

Line 139: Was the CoP displacement time series used in the sample entropy analysis (relative to the velocity or an increment time series)? If so, please state that here.

Answer:

To clarify this issue the following sentence in the section of methods was added: “The CoP displacement increment time series was used to in the sample entropy analysis”.

Reviewer comment

DISCUSSION Line 222: Please add a sentence at the end of this paragraph that states whether your hypotheses were supported and to what extend (i.e., generally, partially, not at all).

Answer:

Thank you for this suggestion. We have corrected this paragraph and added following sentences: “The aim of the present study was to investigate whether an internal and external focus of attention could promote greater changes in the postural regularity of COP. We also aimed to explore how the process of postural control differs between groups of different levels of expertise. Our results supported the hypothesis that biathletes would show a more regular COP signal compared to other groups. However, the second hypothesis could not be supported, because EF did not induce the biggest changes in the automaticity of postural control compared to IF.”

Reviewer comment

Line 287: An additional limitation should be considered. The “aiming to a target” condition was essentially a dual-task condition, so it would have been ideal to also record performance on target aim in conjunction with postural sway. While you may not have these data, it would be good to identify has a limitation or at least a future direction.

Answer:

We completely agree with reviewer, and as it was correctly guessed, we do not have such data. The reviewers suggestion was included in the following sentence in the limitations paragraph: “Third, the “aiming to a target” condition was essentially a dual-task condition, however, we did not record performance on target aim in conjunction with postural sway. We recommend that future studies should include the recording how long the subject has been in the target.”

Reviewer comment

General comment: The discussion primarily focuses on how the biathletes differed from the other groups, which is appropriate given the research question. However, it would be helpful to add text to describe/interpret the findings of the beginners relative to the control group. Significant differences are reported between those groups in Figures 2 and 3, so describing why those differences may have occurred would be important here.

Answer: 

Thank you for this comment, however we did not obtained significant differences between the beginners and control group. In the figures 2 and 3 we ticked the differences between group as a thin dotted line. Whereas the differences between conditions among specific group were marked as a thin solid line. To clarify the figures, we have provided an additional legend.

Fig. 2. The mean values of SampEn in the anterior-posterior direction (standard deviations marked as error bars) in the biathletes, beginners and control group.

Legends:

a thin dotted line: significant differences between groups 

a thin solid line: significant differences between conditions among groups.

Fig. 3. The mean values of SampEn in the medio-lateral direction (standard deviations marked as error bars) in the biathletes, beginners and control group.

Legends:

a thin dotted line: significant differences between groups 

a thin solid line: significant differences between conditions among groups.

Reviewer #2

The authors examine complexity changes within a specific population group that has extensive balance experience relative to a shooting/aiming task and make comparisons of balance performance to a novice and control group. Using sample entropy the COP complexity is indexed for signal regularity and quantified for the 3 different conditions (quiet standing, shooting, and aiming). The main findings show that the biathletes exhibit different COP complexity than the novice and control groups. Within the current form of the manuscript, the authors have an appropriate experimental design and sample to address the main question. However, there lacks adequate rationale and clarity on the background theory and evidence related to this question. The authors need to make significant improvements around the main concepts (balance experience, complexity, attentional focus) and provide clear links of these ideas to their main question/variables.

Answer: 

First of all, we would like to thank the reviewer for a very insightful review. The manuscript was amended thoroughly according to the reviewer’s comments and suggestions. Some issues have risen after reading the review, which is explained in detail below.

Reviewer comment

Title – this does not appear to align with information presented within the manuscript and the concepts provided in the introduction. Closed-looped strategy is not explained/described anywhere within the manuscript. Please address.

Answer: 

We fully understand the reviewer’s concern. After deep consideration, we did not directly examine open and closed strategy. Analysis of this strategy, based on sample entropy is too far-fetched an interpretation. Therefore , we have change a title of our article as follow: “Biathletes present a repeating patterns of postural control to maintain their balance during shooting”

Reviewer comment

p. 8. Line 12 – the abstract starts with a phrasing that uses “also” in the first sentence. I don’t see how this fits within the first sentence. Also, the term “success” has a quite a few different interpretations/meanings. It would be beneficial to clarify on how the authors interpret this term.

Answer: 

Thank you for this comment. We have removed the word “also” from the abstract. Additionally, we have explained, what does the sport success term mean: “More stable posture might be a key factor for shooting scores.”

Reviewer comment

p. 8 line 16 – I would recommend moving “such as entropy” to after deterministic methods.

Answer: 

The phrase “such as entropy” have been moved as follow: “Because COP measurements are highly irregular and non-stationary, the non-linear deterministic methods, such as entropy are more appropriate for the analysis of COP displacement.”

Reviewer comment

p. 8 line 17 – At the beginning of sentences numbers need to be spelled out as do numbers less than 10.

Answer: 

The sentence has been corrected as follow: “ Eight national-level biathletes, 15 non-athletes who prior to the experiment took part in 3 months of shooting training, and 15 non-athletes with no prior rifle shooting experience.”

Reviewer comment

Data is plural – should read “The data were collected…”

Answer: 

We have corrected the sentence as follow: “The data were collected with the use of a force plate.”

Reviewer comment

The conditions related to internal/external focus should be incorporated into the background information of the abstract.

Answer: 

Thank you for pointing this out, we have added additional information as follow:

“Participants performed three balance tasks in quiet standing, the shooting position (internal focus – they were concentrated on maintaining the correct body position and rifle ), and aiming at the target (external focus – they were concentrated on keeping the laser beam centered on the targets).”

Reviewer comment

Some additional details around the methods would be helpful. This could be achieved by removing the details of the non-significant findings in the results part.

The final sentence includes “closed-looped strategy” – see comment about title as that applies here in the abstract too.

Answer:

Once again, thank you for this comment. As we mentioned before, we have removed “closed-loop strategy” term. Additionally, we have change our final conclusion as follow: “They create repeating patterns (more regular signal for COP) to keep one’s balance during the shooting and aiming to the target positions.” We have also removed non-significant findings in the results part.

Reviewer comment

Introduction 

The opening paragraph covers two main ideas that should be separated into independent paragraphs. First, the role of balance on performance and links to injury risk followed by the explanation of COP measurement. Also, the authors are making a general claim on injury risk in relation to balance/postural control. There needs to be improved rationale on this element (if this is a focus of the paper).

Answer:

We fully understand reviewer’s concern. We did not examine risk of falls, so we have removed redundant information from the Introduction 

Reviewer comment

Line 56-57: The statement “its interpretation is still intangible” is not accurate. Significant work has been done on this front and while it is recognized that there remain limitations to our understanding around complexity. Concrete hypotheses associated with this characteristic of human behavior can be found within the literature. The current manuscript would be greatly strengthened when the authors incorporate these perspectives within the introduction. This also will provide them with clearer interpretations of the findings based on the experimental design. Particularly, given the somewhat bidirectional changes in entropy between the experimental groups it is worthwhile to address in more detail how the balance training/experience in biathletes influences the reduction of COP irregularity.

Answer:

Thank you for this comment. We have provided following changes in the introduction: “Although entropy has been used to examine the complexity of human postural control over the last two decades, there remain limitations to our understanding around complexity”…… “Low values indicate a more regular signal (pathology/disability), and high values are a sign of more chaotic properties of the signal (higher expertise) [14,15]. This way of interpretation can suggest that the postural control system response to an unexpected perturbation has a specific pattern that can be observed by measuring the entropy of the signal [16–18]. Manor et al. [19] noticed, that the degree of complexity associated with postural control system behaviour was correlated with the integrity of involved sensory systems. Somatosensory or/and visual impairments contributed to decreased postural sway complexity, which reflected reduced adaptive capacity of the postural control system.”

Reviewer comment 

Page 11, lines 94-98

These two sentences do not align because one is looking at sway magnitude (which is not used in the methods/results section) while the other predicts COP regularity. Clarify whether you expected directional changes for each type of COP variability metric (linear/nonlinear).

Answer:

We fully understand the reviewer’s concern. We did not assessed postural sway. We have removed following sentence from the introduction: “Recent studies have shown that in sports where balance training is one of the most important aspects, athletes are characterized by lower postural sway”. 

Reviewer comment

New term introduced here that needs context and background information. I am assuming it relates to the constrained action hypothesis but that is because I have familiarity with this perspective whereas other readers may not. Thus, novel readers need more context and explanation on what you mean by automaticity.

Answer: 

Thank you for this comment. We have corrected this paragraph as follows: Roerdink et al. [29] noticed a direct relation between the amount of attention invested in postural control and the regularity of COP fluctuation. More irregular COP fluctuations (as indexed by an increase in sample entropy) may be interpreted as an increase in the efficiency or ‘‘automaticity’’. It is consistent with the “constrained action hypothesis” proposed by Wulf [30], which demonstrates the presence of automaticity of motor control processes when attention is withdrawn from controlling one’s balance. The attention can be experimentally diverted from posture when attentional focus is directed to the effects of one’s movement in the environment (external focus - EF) as compared with when one’s focus is directed to body movements (internal focus - IF). Review report has shown that an external focus (EF) is more beneficial for motor performance and learning (including balance) relative to an internal focus (IF) [30]. EF facilitates movement efficiency by promoting movement organization at a more automatic level, while IF involves more conscious control of effectors and consequently disrupts the automaticity of coordination processes.

Reviewer comment 

COP regularity decreases when attention is experimentally diverted from posture e.g. by introducing an additional task [20].

Did participants wear shoes?

Answer: 

The participants did not wear a shoes. They stood barefoot on the force plate. This information have been added to the description of the test: “Participants performed three balance tasks. They stood barefoot on the force plate”

Reviewer comment

Do you know whether participants fixated on a spot during the shooting position condition? Is there additional details from your instructions or did you conduct a commitment check about their focus? The commitment check would also apply to the aiming condition since I think you are tagging that as the external focus of attention manipulation.

Answer: 

During QS the participants were focused on the point placed 5 m away from them on the wall at their eye level. During SP the participants were asked to focus on the specific biathlon position. They were looking straight ahead. During AT the participants were focusing on the aiming and holding a laser mark inside the target shield. We have supplemented the experimental procedure as follow: “First, they were asked to stand in a comfortable position with their feet approximately at shoulder width distance on a force platform, with their gaze fixated at a reference point located 5m away in front of them, (QS – quiet standing). Next, they were asked to stand while holding a rifle in a standing shooting position without any fixation point in front of them (SP – Shooting position).They were asked to focus on holing shooting posture. In the last trial, participants were given a target at a distance of 5 meters (target size was adjusted for distance). The participants were asked to focus on aiming and holding a laser mark inside the target (AT – Aiming to target). The laser pointer was fixed at the end of the rifle’s barrel.”

Reviewer comment

Line 138-139: the calculation of COP description is not clear. The AMTI force plate will provide a direct output of the COP trajectory without the need for calculation. Can you please further explain this sentence.

Answer:

We do not fully understand the reviewer’s concern. Although, it is possible to obtain COP from the AMTI’s generic software, its’ direct outputs are forces and moment of forces, which we use in the further data processing (i.e. COP calculation). We believe that this very common practice which does not need explanation. 

Reviewer comment

Improvement in the description of the Sample Entropy analysis is needed to help readers not fully familiar with this non-linear metric – particularly, lines 139-142.

Answer:

The description of the Sample Entropy analysis was improved as follows: “Sample entropy is defined as the negative logarithm for the conditional probability that vectors of length m, which are similar in a time series, remain similar for length m +1 (excluding self matches). Calculations were based on the method described by Richman Moorman (2000) [30] using the following formula in Eq (1):

 (1)

Where m represents the number of data points to compare in a data series, r represents tolerance level, and N represents the length of the time-series. 

A is the number of alike vector lengths (m+1) falling within a relative tolerance limit (r times standard deviation of the time series) and B is the number of alike vector lengths (m) falling within the tolerance limit.

A perfectly repeatable time series (with similar distances between data points) would elicited a SampEn value equaling 0 and a perfectly random time series would give a SampEn value converging toward infinity [32]. The values of the input parameters in this experiment were based on the results of Ramdani et al. [11]. Template vector length (m) was set to 3, and tolerance (r) to 0.2. 

Reviewer comment

The authors need to check the degrees of freedom for the ANOVA results. Based on the design, there are 3 groups and 3 tasks with a total of 38 participants included (unless some were omitted but that would need to be added to the manuscript if that was the case), this would lead to different DOF in the F statistic.

Answer:

Thank you for pointing this out. We have made mistake in the description of degrees of freedom, it should be “F (4, 105)”. We had 114 total observation and 9 total interaction, so the value of error should be 105. We have corrected this value on the manuscript. 

Reviewer comment

It would be beneficial to reference figures 2 and 3 earlier in the results section to help draw the reader’s attention to the significant differences. I know the figures will be placed nearby but the first paragraph of the AP and ML results relate very much to the data presented in these figures.

Answer:

Thank you for this comment. We reference figures 2 and 3 earlier, after sentences where we pointed out the significant differences.

Reviewer comment

For the A-P direction results, the authors provide the significant interaction effect but then I am confused by the remaining details in the first paragraph compared to the second. If an interaction effect is present there is no need to follow-up on the main effects (group and condition), but it is hard to follow whether second paragraph (lines 181-193) is the post-hoc analysis of the significant interaction effect.

This also exists in the M-L direction results. Please address

Answer:

We fully understand the reviewer’s concern. Two paragraph present the same interaction effect. However, first presents an intergroup comparison between the three groups and second presents an intragroup comparisons during QS, SP and AT. To clarify, we have added the additional headline in section of results. 

Reviewer comment

p.15 line 177-179 – I think these two sentences need to be merged together do better illustrate that the group differences were in the AT condition.

Answer:

Thank you for this comment We have merged the sentences as follow: ”The same pattern was observed in the AT condition, there was significant difference between the CG and biathletes, and between beginners and biathletes, p<0.05

Reviewer comment

p. 15 line 188-189: How do you know there was a small change between consecutive trials when the data were averaged across the trials as stated in the statistical analysis section? Please clarify and/or adjust the manuscript.

Answer:

We fully understand the reviewer’s concern. Terms “consecutive trials” refers to consecutive conditions. We have corrected the sentence as follow: “In the biathletes group, the changes between consecutive conditions were much smaller and, contrary to both other groups’ values in the SP and AT trials, were lower than in the QS condition.”

Reviewer comment

p.17, line 220 – attentional focus has not been discussed or included in the original hypotheses. If you are going to interpret the findings relative to this concept additional background information is needed in the introduction.

Answer:

Thank you for pointing this out. We have added additional background information in the introduction about attentional focus, see lines 83- 95.

Reviewer comment

p. 17, line 223-231 – There appears to be some inconsistencies in the information detailed in this paragraph. The first sentence states no differences between groups, followed by that being in line with other studies. But, then it states increased entropy for athletes. Please clarify.

Answer:

Thank you for pointing this out. However, the first paragraph refers to QS standing position, as a simple motor task. In turn, the second paragraph describes changes in the values of sample entropy in specific posture position like SP and AT.

Reviewer comment

p. 18, line 245-246 – The reference to the findings being in line with the “constrained action hypothesis” needs further elaboration. In what way do the results support the many elements of this perspective?

Answer: 

Thank you for this comment. We have changed this paragraph as follow: “These outcomes are contrary to the CG and BG results, which are in line with the “constrained action hypothesis.” For BG and especially for CG, SP and AT were not familiar position. In contrast to Biathletes, in these motor tasks, we could distinguish both primary (balance) and secondary task (IF and EF). Introducing additional focus in BG and CG experimentally withdrawn attention from the balance task. What was visible by higher values of sample entropy, which might correspond with the more automatic process of the primary motor task. According to Rhae et al. [28] EF provide bigger changes in values of sample entropy compared to IF. However, in our study there were no differences between IF and EF. The two types of additional focus included in this research triggered the same changes in COP structure, which does not confirm our second research hypothesis.” 

Reviewer comment

p. 18, line 263-264 – the interpretation of close-loop (should be closed-loop) pattern needs further explanation on why the authors believe this to be the case within the results. Plus, as with previous comments around this concept, a rationale for its use is needed in the introduction.

Answer: 

Once again, thank you for this comment. As we mentioned before have resigned to interpreted our results based on closed-loop strategy. So we have removed following sentences: “. From this perspective, we can say that biathletes’ posture is controlled mostly in a close-loop pattern even though shooting is a single short non-cyclic movement”.

Reviewer comment

p. 19, line 282-286 – depends on the task demands, which authors are correct here in terms of reducing COP variability irregularity but other tasks may not follow the same pattern.

Also, task success in shooting is dependent on the barrel variability and may not be as related to the movement/body variations. Classic work by Arutyunyan showed this characteristic.

Answer: 

We have added this information to this paragraph as follow: “It's worth pointing out, that success in shooting is dependent on barrel variability and may not be as related to the body oscillations.” 

Reviewer comment

Table 1 needs a period instead of a comma under the Biathlon data.

Answer:

We have changed comma for dot. 

Reviewer comment

p. 16, line 199 – there is only one interaction so it should be used in the plural form – should read “significant interaction.”

Answer:

We have changed sentence as follow: “Similar to the AP plane, there was statistically significant interaction between experimental groups and conditions, F(4, 70)=4.525, p<0.05, ηG2=0.131.”

Reviewer comment

p. 18, line 254 – please address this sentence for clarity purposes.

Answer: 

The references is placed on the next line as “’[37]” – Lubetzky A V., Price R, Ciol MA, Kelly VE, McCoy SW. Relationship of multiscale entropy to task difficulty and sway velocity in healthy young adults. Somatosens Mot Res. 2015;32(4):211–8.

Reviewer comment

p.19, line275 – biathletes should be in lower case for consistency within the manuscript.

Answer: 

We have corrected word biathletes as follow: “Based on the current research, biathletes are characterized by more regular signal of COP.”

---

## [Decision Letter · Decision Letter 1]

2 Mar 2022

PONE-D-21-34186R1Biathletes present a repeating patterns of postural control to maintain their balance during shootingPLOS ONE

Dear Dr. Michalska,

Thank you for submitting your manuscript to PLOS ONE. After careful consideration, we feel that it has merit but does not fully meet PLOS ONE’s publication criteria as it currently stands. Therefore, we invite you to submit a revised version of the manuscript that addresses the points raised during the review process.

As you will see, both Reviewers feel that your revisions are quite strong. However, Reviewer 2 has identified some lingering problems with the text in the Results section. The value of your contribution depends powerfully on the clarity of the Results section: Readers must know exactly what the results are. So, I ask you to revise the text, as recommended by the Reviewer.

We look forward to receiving your revised manuscript.

Kind regards,

Thomas A Stoffregen, PhD

Academic Editor

PLOS ONE

Journal Requirements:

Reviewers' comments:

Reviewer's Responses to Questions

**Comments to the Author**

1. If the authors have adequately addressed your comments raised in a previous round of review and you feel that this manuscript is now acceptable for publication, you may indicate that here to bypass the “Comments to the Author” section, enter your conflict of interest statement in the “Confidential to Editor” section, and submit your "Accept" recommendation.

Reviewer #1: All comments have been addressed

Reviewer #2: All comments have been addressed

2. Is the manuscript technically sound, and do the data support the conclusions?

Reviewer #1: (No Response)

Reviewer #2: Partly

3. Has the statistical analysis been performed appropriately and rigorously? 

Reviewer #1: (No Response)

Reviewer #2: Yes

4. Have the authors made all data underlying the findings in their manuscript fully available?

Reviewer #1: (No Response)

Reviewer #2: Yes

5. Is the manuscript presented in an intelligible fashion and written in standard English?

Reviewer #1: (No Response)

Reviewer #2: No

6. Review Comments to the Author

Reviewer #1: (No Response)

Reviewer #2: The manuscript is much improved from its initial submission. The introduction is more coherent and on point with the research question. Details in the methods have been clarified/modified to enhance the readers understanding of the design. However, the results are still written in an incomplete manner.

The first sentence (line 182-183) has no dependent variable provided in it or is it in the heading, so it is unclear as to what this finding is relative to. The same pretty much happens in the second paragraph but has SampEn referenced in the middle rather than at the beginning.

For the “intragroup comparisons during QS, SP, and AT” section, was the significant finding the main effect of condition or the interaction between condition and group? Some of the results provided here do not make it clear. Some language seems to reference the condition effect while other parts speak about group differences being present. Improvement and clarity needed.

Similar issues are present for the ML direction material.

The discussion is much improved and matches the introduction in a better manner with proper interpretations.

7. PLOS authors have the option to publish the peer review history of their article (what does this mean?). If published, this will include your full peer review and any attached files.

Reviewer #1: **Yes: **Christopher K. Rhea

Reviewer #2: No

---

## [Author Response · Author response to Decision Letter 1]

22 Mar 2022

Reviewer 1

We are very grateful for a such an approving review and acceptance for publication. It strongly motivates us to conduct further research.

Reviewer 2 

The manuscript is much improved from its initial submission. The introduction is more coherent and on point with the research question. Details in the methods have been clarified/modified to enhance the reader’s understanding of the design. However, the results are still written in an incomplete manner. The discussion is much improved and matches the introduction in a better manner with proper interpretations.

Answer:

We appreciate this comment. We have provided all necessary changes and corrections to clarify and improve the rationale of our research. We are aware that the previous results section might be misleading and unclear. Therefore, we have revised the entire section, see lines 180-225.

Reviewer comment:

The first sentence (line 182-183) has no dependent variable provided in it or is it in the heading, so it is unclear as to what this finding is relative to. The same pretty much happens in the second paragraph but has SampEn referenced in the middle rather than at the beginning.

Answer:

Thank you for this comment. We have provided a dependent variable and have changed the “AP” and “ML” headings in “Sample entropy in the anterior-posterior plane” and “Sample entropy in the medio-lateral plane,” respectively.

Reviewer comment:

For the "intragroup comparisons during QS, SP, and AT" section, was the significant finding the main effect of condition or the interaction between condition and group? Some of the results provided here do not make it clear. Some language seems to reference the condition effect while other parts speak about group differences being present. Improvement and clarity needed. Similar issues are present for the ML direction material.

Answer:

We fully understand the reviewer’s concern. As previously mentioned, we have revised this section, see lines 180-225.

We have also sent our manuscript to Professional English Services. All grammatical corrections have been highlighted in the Revised manuscript with track changes

---

## [Decision Letter · Decision Letter 2]

4 Apr 2022

Biathletes present repeating patterns of postural control to maintain their balance while shooting

PONE-D-21-34186R2

Dear Dr. Michalska,

We’re pleased to inform you that your manuscript has been judged scientifically suitable for publication and will be formally accepted for publication once it meets all outstanding technical requirements.

Kind regards,

Peter Andreas Federolf

Academic Editor

PLOS ONE

Reviewers' comments:

Reviewer's Responses to Questions

**Comments to the Author**

1. If the authors have adequately addressed your comments raised in a previous round of review and you feel that this manuscript is now acceptable for publication, you may indicate that here to bypass the “Comments to the Author” section, enter your conflict of interest statement in the “Confidential to Editor” section, and submit your "Accept" recommendation.

Reviewer #1: All comments have been addressed

Reviewer #2: All comments have been addressed

2. Is the manuscript technically sound, and do the data support the conclusions?

Reviewer #1: (No Response)

Reviewer #2: Yes

3. Has the statistical analysis been performed appropriately and rigorously? 

Reviewer #1: (No Response)

Reviewer #2: Yes

4. Have the authors made all data underlying the findings in their manuscript fully available?

Reviewer #1: (No Response)

Reviewer #2: Yes

5. Is the manuscript presented in an intelligible fashion and written in standard English?

Reviewer #1: (No Response)

Reviewer #2: Yes

6. Review Comments to the Author

Reviewer #1: (No Response)

Reviewer #2: (No Response)

7. PLOS authors have the option to publish the peer review history of their article (what does this mean?). If published, this will include your full peer review and any attached files.

Reviewer #1: No

Reviewer #2: No

---

## [Editor Report · Acceptance letter]

18 Apr 2022

PONE-D-21-34186R2 

Biathletes present repeating patterns of postural control to maintain their balance while shooting 

Dear Dr. Michalska:

I'm pleased to inform you that your manuscript has been deemed suitable for publication in PLOS ONE. Congratulations! Your manuscript is now with our production department. 

Kind regards, 

on behalf of

Dr. Peter Andreas Federolf 

Academic Editor

PLOS ONE